# Distinct spatiotemporal mechanisms underlie extra-classical receptive field modulation in macaque V1 microcircuits

Christopher A Henry[1,2]*, Mehrdad Jazayeri[3], Robert M Shapley[1], Michael J Hawken[1]*

[1]Center for Neural Science, New York University, New York, United States; [2]Dominick Purpura Department of Neuroscience, Albert Einstein College of Medicine, Bronx, United States; [3]Department of Brain and Cognitive Sciences, McGovern Institute for Brain Research, Massachusetts Institute of Technology, Cambridge, United States

**Abstract** Complex scene perception depends upon the interaction between signals from the classical receptive field (CRF) and the extra-classical receptive field (eCRF) in primary visual cortex (V1) neurons. Although much is known about V1 eCRF properties, we do not yet know how the underlying mechanisms map onto the cortical microcircuit. We probed the spatio-temporal dynamics of eCRF modulation using a reverse correlation paradigm, and found three principal eCRF mechanisms: tuned-facilitation, untuned-suppression, and tuned-suppression. Each mechanism had a distinct timing and spatial profile. Laminar analysis showed that the timing, orientation-tuning, and strength of eCRF mechanisms had distinct signatures within magnocellular and parvocellular processing streams in the V1 microcircuit. The existence of multiple eCRF mechanisms provides new insights into how V1 responds to spatial context. Modeling revealed that the differences in timing and scale of these mechanisms predicted distinct patterns of net modulation, reconciling many previous disparate physiological and psychophysical findings.

*For correspondence:
christopher.henry@einstein.yu.edu (CAH);
michael.hawken@nyu.edu (MJH)

**Competing interests:** The authors declare that no competing interests exist.

## Introduction

Object vision relies on integrating and differentiating local image features to form a representation of the visual input. Many low- to mid-level computations emerge in neuronal response properties in primary visual cortex (V1) where neurons have both classical (CRF; *Hubel and Wiesel, 1962*; *Hubel and Wiesel, 1968*; *Angelucci et al., 2002*) and extra-classical (eCRF; *Allman et al., 1985*; *Levitt and Lund, 1997*; *Angelucci et al., 2002*) receptive fields. The CRF provides the spatio-temporal filtering properties of the neuron, and consists of regions where stimuli directly evoke spiking activity. The eCRF modulates CRF spiking responses, providing contextual components that are especially important in complex visual recognition (*Meese et al., 2009*; *Meese and Baker, 2013*), as well as assignment of border-ownership that is crucial to figure-ground perception (*Lamme, 1995*; *Kogo and Wagemans, 2013*; *Russell et al., 2014*).

Many issues about the eCRF mechanisms and their computations are unresolved. There is considerable debate as to what extent the eCRF produces facilitation or suppression (*Angelucci et al., 2017*), and thus whether it involves feature integration or differentiation. Furthermore, it remains unclear whether eCRF modulation arises from a single or multiple mechanisms. Previous studies have also found timing-differences of eCRF modulation (*Müller et al., 2003*; *Smith, 2006*; *Henry et al., 2013*); understanding stimulus-dependent signal-timing is fundamental for determining how the eCRF gates CRF responses. Further, the extent to which the eCRF mechanisms are inherited or emerge via computations within the cortical microcircuit is unclear. In addition, there is recent

interest in how the magnocellular and parvocellular streams contribute to contextual processing in V1 and extra-striate cortex (*Jones et al., 2012*; *Henry et al., 2013*; *Conway, 2014*; *Klauke and Wachtler, 2015*). How eCRF properties differ across the laminar architecture in V1 may strongly impact how context information is relayed along distinct cortical output pathways.

In this paper, we introduce the study of eCRF contextual modulation in the temporal domain via reverse correlation. This approach enabled separate characterization of multiple functional mechanisms within the eCRF because facilitation and suppression exhibited distinct spatial and temporal dependencies. Furthermore, we studied the dynamics of eCRF modulation across cortical layers to assess whether these response dynamics were distributed or elaborated in distinct feedforward and feedback output pathways.

Our new findings provide evidence for multiple dynamic mechanisms in the eCRF (*Ringach et al., 2003*; *Xing et al., 2005*) that have distinct spatial profiles and orientation tuning: 1) orientation-tuned-facilitation was found within the CRF and near the CRF/eCRF border but was absent at larger spatial extents; 2) orientation-untuned suppression was localized near the CRF/eCRF border; 3) orientation-tuned suppression was found at larger spatial extents than facilitation or untuned suppression. Different eCRF mechanisms were found in different cortical layers and their laminar location was related to parallel processing in magnocellular and parvocellular streams that contribute differentially to the dorsal (*Goodale, 2014*) and ventral (*Kravitz et al., 2013*) extra-striate processing streams. On the basis of our experimental findings, we propose that there are various eCRF mechanisms that are activated by different spatial and temporal stimulus configurations and by different feedforward input streams. Understanding when and where these mechanisms operate can reconcile previous conflicting interpretations of eCRF function.

## Results

The time course of eCRF modulation was measured and its dependence upon orientation, spatial phase, and spatial extent determined using a reverse-correlation stimulation paradigm (*Figure 1*) in 106 V1 neurons (37 simple, 69 complex cells). Neurons were recorded across all cortical layers, and assigned to a layer following subsequent histological reconstruction (see 'Materials and methods'). Only neurons with well-isolated spike waveforms that remained stable for the duration of the experiments were included in the study. Their receptive fields were parafoveal, with centers ranging from 1 to 6 degrees eccentricity in the visual field. Across the population, selectivity for stimulus orientation, as measured by circular variance of the tuning curve, was diverse and ranged from 0.02 to 0.98 (median: 0.59), consistent with prior studies (*Ringach et al., 2002*). Tuning for preferred stimulus size, measured as the aperture radius of an optimal stimulus, also spanned a wide range with optimal sizes ranging from 0.12 to 4 degrees (median: 0.45 degrees), which is in agreement with distributions reported in a prior study (*Sceniak et al., 2001*). In this results section, we first describe the components of eCRF modulation seen across the population and then investigate how these response dynamics varied with both the spatial scale of the stimuli and with neurons' laminar locations. Finally, modeling results demonstrate the net effects that multiple eCRF mechanisms have for specific contextual stimuli.

### Three components of eCRF modulation

The CRF stimulus was an achromatic drifting grating of the individual neuron's preferred orientation, spatial and temporal frequency, and drift direction. The CRF grating moved continuously, evoking a steady spike discharge. For each neuron, the CRF stimulus contrast was set to evoke ~50% of the neuron's maximum response, ensuring that the neuron's drive provided sufficient dynamic range to observe both suppression and facilitation via eCRF stimulation. The timing of eCRF influence was probed by briefly presenting additional high-contrast achromatic drifting gratings in the eCRF that changed randomly in orientation and direction every two stimulus frames (20 ms). The eCRF stimulus sequence contained occasional stimuli of 0% luminance contrast (mean grey) to provide a reference response with no eCRF stimulation (*Figure 1A*). Analysis consisted of reverse correlation of action potentials (spikes) with preceding eCRF stimuli. This reverse correlation approach is similar to those used to study the generation of CRF selectivity for orientation and spatial frequency (*Ringach et al., 1997*; *Bredfeldt and Ringach, 2002*; *Xing et al., 2005*) and binocular interactions (*Tanabe et al., 2011*; *Tanabe and Cumming, 2014*).

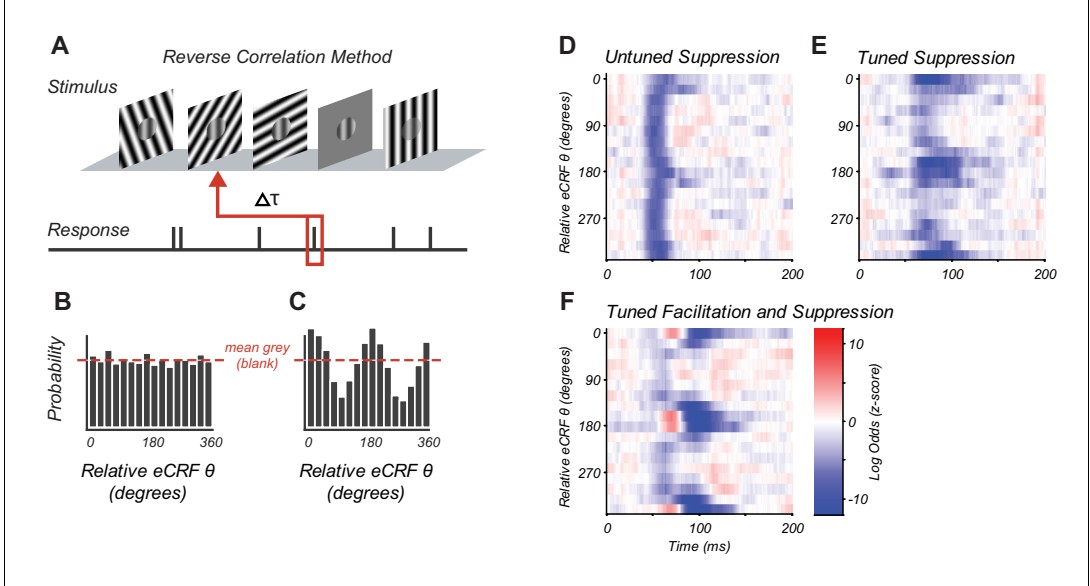

**Figure 1.** Reverse correlation approach and example eCRF dynamics. (**A**) Orientations were briefly and randomly presented in the eCRF alongside optimal CRF stimulation. The probability of an eCRF stimulus prior to a spike was calculated for each delay (τ); blank eCRF stimuli were included as controls. (**B**) A hypothetical probability distribution for τ = 0 as a function of eCRF orientation (relative to CRF preference). The red dashed line shows the probability associated with the mean grey (blank) stimulus and no modulatory effect. (**C**) Same as panel (**B**) but for a delay τ where there is a strong facilitation at the preferred orientation (0, 180 deg) and suppression at the orthogonal orientation (90, 270 deg). (**D–F**) False color maps showing the log odds ratio (of the probability of each eCRF stimulus compared to blank) over time. Red indicates facilitation and blue suppression. (**D**) An example neuron showing strong suppression at all orientations (untuned) beginning at around 50 ms. (**E**) An example neuron showing strong suppressive modulation at its preferred orientation (tuned) around 75–100 ms. (**F**) An example neuron showing both early tuned collinear facilitation (red at ~60 ms) followed by delayed tuned suppression (blue at 90–140 ms).

The online version of this article includes the following source data for figure 1:

**Source data 1.** Source data for *Figure 1*.

For a fixed time lag, the modulation caused by a given eCRF stimulus was quantified as the log ratio of the probability that a given eCRF stimulus occurred prior to a spike over the probability that a blank eCRF stimulus occurred.

$$R_\theta(\tau) = \log\left(\frac{p(\theta|\tau)}{p(blank\,|\tau)}\right)$$

We refer to the function $R_\theta(\tau)$ as the log-odds ratio (LOR), where $\tau$ is the time preceding a spike and $\theta$ is the orientation of the eCRF stimulus. Positive values (e.g. *Figure 1C* at 0, 180 degrees) indicate eCRF stimulus-induced increases in spiking (interpreted as response facilitation). Negative values (e.g. *Figure 1C* at 90, 270 degrees) indicate eCRF stimulus-induced decreases in spiking (interpreted as response suppression). Values near zero (*Figure 1B*) indicate no difference between eCRF stimulus and blank (interpreted as no response modulation). *Figure 1* and later figures plot the LOR as a function of orientation and time as a false color map with facilitation shown in red and suppression in blue.

We observed three main response components within the population, as illustrated by three example V1 neurons (*Figure 1D–F*). All eCRF orientations are plotted relative to the neuron's CRF orientation preference. Some neurons showed an early tuned facilitation (*Figure 1F*, red) often followed by a distinct, tuned suppression (*Figure 1F*, blue). Many neurons showed early response suppression that was equal across all eCRF orientations (*Figure 1D*: blue band at 50–70 ms). Such untuned suppression was often followed by orientation-tuned-suppression for stimuli collinear with the CRF stimulus (*Figure 1D*: 75–100 ms). In many neurons, this delayed orientation-tuned suppression was predominant (*Figure 1E*). The magnitudes of the facilitation and suppression were calculated as the z-scores of the LOR compared to the baseline variance (see 'Materials and methods'). Neurons with z-scores > 2, that is deviation from the baseline response ±2 s.d., were deemed to

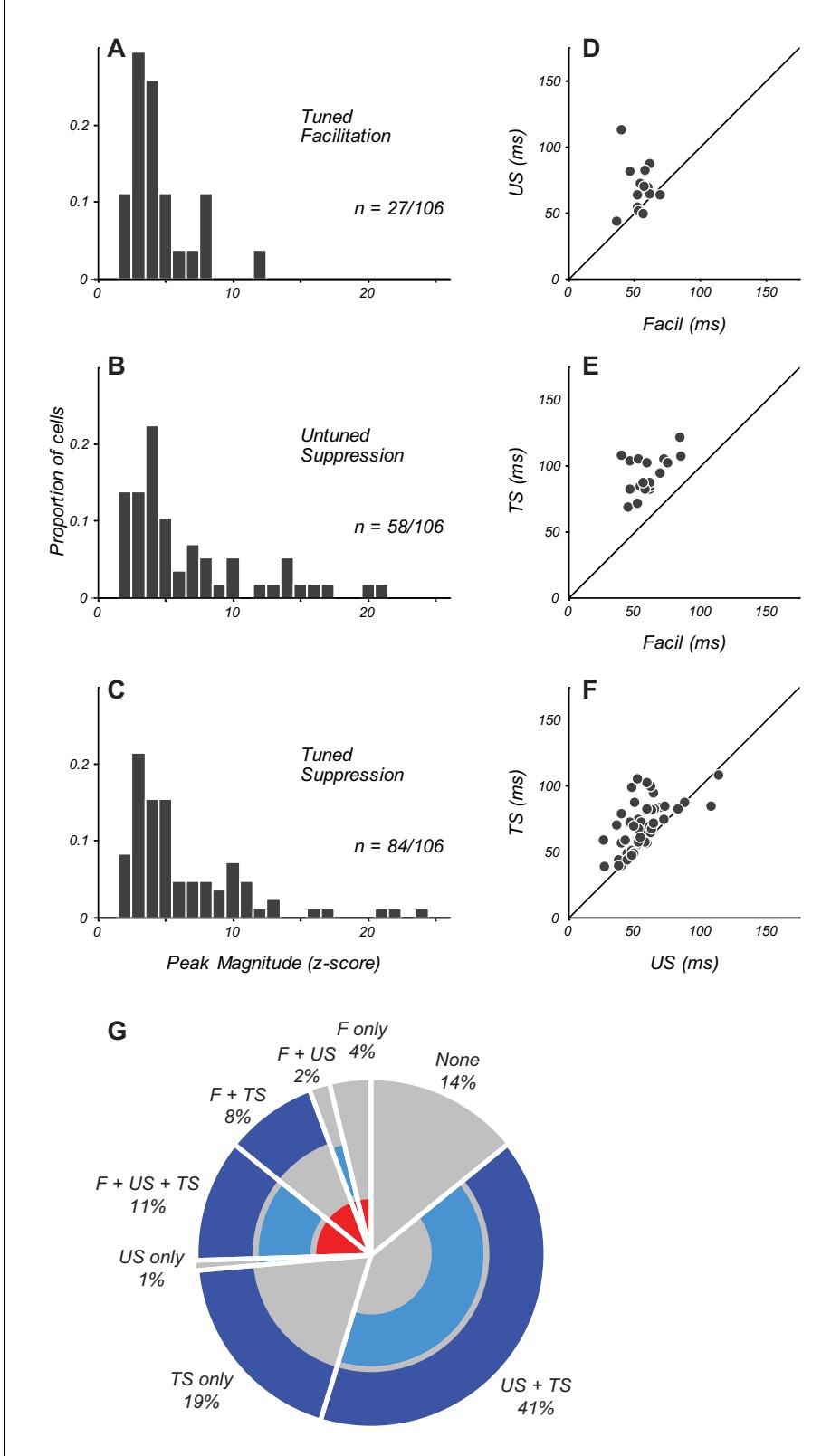

**Figure 2.** Magnitude and timing of eCRF components. (A–C) The distribution of peak modulation strength in neurons with significant eCRF components of tuned facilitation (A), untuned suppression (B), and tuned suppression (C). Neurons with z-scored magnitudes > 2 are included. (D) Comparison in individual neurons of the peak times of facilitation and untuned suppression (US). Diagonal indicates unity line. (E) Timing of facilitation and tuned suppression (TS). (F) Timing of untuned and tuned suppression. (G) Percentage of neurons in the population exhibiting specific combinations

*Figure 2 continued on next page*

*Figure 2 continued*

of significant eCRF components. Facilitation (labeled F) is indicated by the inner red ring. Untuned and tuned suppression indicated by middle and outer blue rings (US and TS, respectively). The absence of significant eCRF modulation is shown in grey.

The online version of this article includes the following source data for figure 2:

**Source data 1.** Source data for *Figure 2*.

have a significant component of eCRF modulation. In the following section, we examine how the magnitude and timing of eCRF modulation varied across our V1 population.

## Magnitude of eCRF facilitation and suppression across the V1 population

Overall, suppressive eCRF components were more prevalent than eCRF facilitation, and also had larger peak modulation amplitudes. Only 25% of neurons (27/106; *Figure 2A*) had significant early facilitation from eCRF orientations collinear to the CRF grating. More than half of the population (55%, 58/106; *Figure 2B*) had significant untuned suppression (estimated from the LOR value at orthogonal-to-preferred eCRF orientations). Significant tuned suppression at collinear orientations was observed in 79% of neurons (84/106; *Figure 2C*). All three components of eCRF modulation could be commingled in individual neurons' responses. A small number of neurons showed both an early eCRF facilitation followed by either untuned suppression (14/106) or tuned suppression (21/106). For those neurons that had significant components of eCRF modulation, the average peak z-scores were 4.6 ± 0.5 for tuned facilitation, 6.6 ± 0.6 for untuned suppression, and 6.4 ± 0.5 for tuned suppression (mean ± s.e.m.), indicating that suppressive mechanisms had greater strength as well as prevalence across our population.

In individual neurons, the distinct eCRF components were present in varying combinations. In the population of 106 neurons, only 15 neurons showed no significant eCRF components at all; this group of neurons lacking eCRF modulation was distributed across layers 2/3, 5, and 6, and were largely absent from layers 4b and 4C. The other 91 neurons in the sample had various combinations of facilitation, untuned suppression, and tuned suppression from the eCRF.

There were 27/91 neurons that showed facilitation. Of these, four had only facilitation present, whereas in two, facilitation co-occurred with untuned suppression only, and in nine, it co-occurred with tuned suppression only. The remaining 12 neurons had facilitation co-occurring alongside both untuned and tuned suppression. In 64/91 neurons lacking facilitation, one had only untuned suppression present, 20 had only tuned suppression present, and 43 had both untuned and tuned suppression co-occurring. The pie chart in *Figure 2G* provides a summary visualization of the percentage of neurons that exhibited each of these specific combinations of eCRF components.

## Timing of eCRF facilitation and suppression

The precise timing of eCRF components provides key information about underlying mechanisms. We used the time to peak modulation as a metric to compare the dynamics between component eCRF mechanisms. For the 25% of neurons that showed eCRF facilitation, the peak time was 60 ± 3 ms (n = 27, mean ± s.e.m.). The average peak time for untuned suppression was similar (55 ± 2 ms; n = 58). When untuned suppression and facilitation were present within the same neurons, the time of peak facilitation was significantly earlier than that of suppression (*Figure 2D*, n = 14, p<0.005), although for several neurons the times were similar. The average peak time of tuned suppression (76 ± 3 ms) was significantly later than that of facilitation (*Figure 2E*, p<0.0001) and untuned suppression (*Figure 2F*, p<0.0001, all paired tests are Wilcoxon signed-rank tests unless otherwise indicated). We further compared eCRF component dynamics by measuring the length of time over which we observed a significant response. The average duration of the three components were 16 ± 2 ms for facilitation, 23 ± 2 ms for untuned suppression, and 36 ± 3 ms for tuned suppression. Thus tuned suppression, in addition to being the most prevalent eCRF mechanism, also exhibited the most prolonged temporal profile.

## Spatial extent of eCRF mechanisms

The border between the CRF and eCRF was defined by the closest regions to the CRF in which stimulation produced no evoked spiking response when there was no stimulus in the CRF. The results above reflect experiments in which the inner extent of the dynamic probe stimulus was placed at the eCRF border. In additional experiments, we varied the position of the dynamic stimulus to probe the spatial extent of the component mechanisms that underlie eCRF modulation. *Figure 3* illustrates the three spatial configurations employed. Two spatial configurations (*Figure 3D, F*) were added, to compare to the effects from stimuli that began at the eCRF border (*Figure 3E*). In

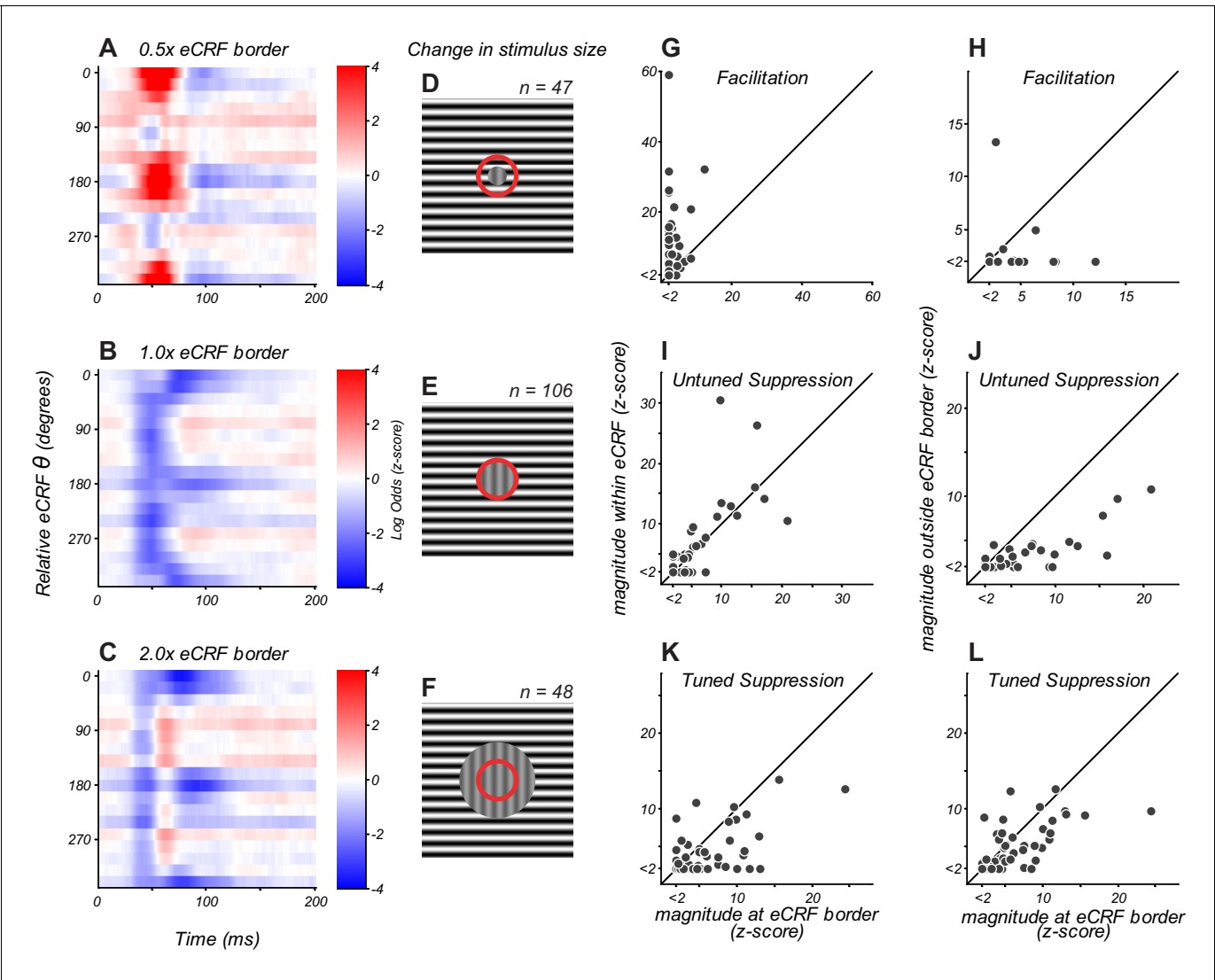

**Figure 3.** eCRF dynamics change with spatial configuration. (A–C) eCRF modulation over time as a function of relative eCRF orientation for three spatial configurations, averaged over neurons. The location of the inner diameter of the surround stimulus is indicated at the top of each map. The color scales indicate modulation (red, facilitation; blue, suppression). (D–F) Spatial configurations associated with the responses in panels (A–C). The red circle denotes the border between the CRF and eCRF. Vertical grating indicates stimulus driving the CRF. Horizontal grating indicates where the dynamic eCRF stimulus was shown. (G–L) Change in the magnitude of modulation with spatial configuration, for eCRF components of facilitation (G ,H), untuned suppression (I, J), and tuned suppression (K, L).

The online version of this article includes the following source data and figure supplement(s) for figure 3:

**Source data 1.** Source data for *Figure 3*.

**Figure supplement 1.** Time course of facilitation and suppression at different spatial phases of the eCRF stimulus.

**Figure supplement 1—source data 1.** Source data for *Figure 3—figure supplement 1*.

*Figure 3D–F*, the red circle represents the CRF/eCRF border, the vertical grating indicates the stimulus driving the CRF, and the horizontal grating represents the region in which the eCRF stimulus sequence was shown.

We compared the dynamics of eCRF modulation for the three spatial conditions tested (*Figure 3A–C*) by averaging across all neurons. As neurons with strong modulation could bias this measure, we confirmed that results were similar when neurons were weighted equally by normalizing the maps of each neuron before averaging (not shown). In the following sections, we report how the strength of eCRF components varied with spatial configuration, on average and within individual neurons.

## Results at 1 x eCRF

The average eCRF dynamics with the inner diameter of the probe stimulus at the CRF/eCRF border are shown in *Figure 3B*. There was strong untuned suppression at 40–60 ms, with strong tuned suppression appearing at 80–120 ms. There was response facilitation for some individual neurons (*Figure 3G*, x-axis), but this was not evident in the average response since such facilitation was relatively weak and had temporal overlap with early untuned suppression.

## 0.5 x compared to 1 x eCRF

By placing the inner edge of the probe stimulus at 0.5 x the eCRF border (*Figure 3D*), we could infer how modulation dynamics change by engaging regions closer to the CRF. In this configuration, early facilitation became more prevalent (*Figure 3A*), with 91% (43/47) of neurons showing significant facilitation (Figure 3G, y-axis). The magnitude of facilitation was significantly greater in individual neurons than it was at the 1 x eCRF border condition (*Figure 3G*, p<0.0001). On average, facilitation peaked around 55 ms and was strongly tuned to orientations that were collinear with neurons' preferred orientation (*Figure 3A*). For those neurons that exhibited facilitation at the eCRF border, there was no significant difference in the time of peak facilitation when probed with the closer 0.5 x eCRF border condition (p=0.61, paired t-test). However, as expected, there was a significant increase in the peak magnitude of facilitation in the 0.5 x eCRF border condition (p=0.005). Together, these two results suggest that the facilitative signals from near regions of the eCRF were most probably driven by the spatial continuation of central CRF mechanisms.

Untuned and tuned suppression remained present with this closer eCRF configuration and maintained the same response dynamics (*Figure 3A*). The magnitude of untuned suppression was not significantly different between the 0.5x and 1x border conditions (*Figure 3I*, p=0.15), indicating that untuned suppression comes mainly from signals evoked by visual stimulation beyond 1 x eCRF.

Tuned suppression showed a significant correlation between the two spatial conditions (*Figure 3K*, Pearson's correlation, r = 0.54, p=0.0001), though it was somewhat reduced at 0.5 x eCRF (*Figure 3K*). This apparent reduction in magnitude probably resulted from the temporal overlap with strong tuned facilitation evoked within the CRF, leading to partial cancellation of overall suppression strength.

## 1 x compared with 2 x eCRF

By comparing response dynamics from 1 x eCRF border with those evoked beyond 2 x eCRF border, we could infer what modulation arises from just outside the eCRF border. With the distant stimulus, early facilitation was absent on average (*Figure 3C*), was observed in only 4/47 individual neurons, and was significantly weaker than that at the eCRF border (*Figure 3H*, p<0.02). These results indicate that early facilitation arises from extensions of local CRF mechanisms.

Untuned suppression on average was considerably weaker when evoked by distant stimuli (*Figure 3C*): it was present in 19/47 neurons and had significantly lower magnitude than that from at the eCRF border (*Figure 3J*, p<0.0001). This result implies that a majority of untuned suppression emerged from the annular region between 1x and 2x the eCRF inner border.

By contrast, tuned suppression remained clearly evoked on average by distant stimuli (*Figure 3C*). For individual neurons, the magnitude of tuned suppression was significantly correlated with that measured at the 1 x eCRF border condition (*Figure 3L*, Pearson's correlation, r = 0.63, p<0.0001). This suggests that tuned suppression was evoked mainly from regions beyond 2 x eCRF in most V1 neurons.

## Spatial phase dependence of eCRF mechanisms

To examine spatial phase dependence, eCRF stimuli were presented at two different phases relative to the grating in the CRF (in-phase or 180 degrees out-of-phase); all analyses thus far ignored, and thus marginalized over, spatial phase differences. We separately determined how eCRF modulation from collinear stimuli depended jointly upon the spatial extent and the relative spatial phase of the surround stimuli. Results are given in *Figure 3—figure supplement 1*. To summarize, our results suggested that there was no phase-dependence of either component of suppression but there was spatial phase-dependence of tuned facilitation, seen only in simple cells. These results provide further evidence that the tuned facilitation from the eCRF activated the edge of a central CRF mechanism.

## Laminar organization of eCRF modulation

In primate V1 there are distinct laminar differences in response latencies (*Maunsell and Gibson, 1992*; *Nowak et al., 1995*) and in the amount of eCRF suppression (*Sceniak et al., 2001*). For 81 neurons assigned to layers (see 'Materials and methods'), we asked how the spatiotemporal dynamics and component mechanisms of eCRF modulation covaried with laminar location.

Orientation-tuned suppression was evident in both the magnocellular input layer 4Cα and the parvocellular input layer 4Cβ. However, the timing was very different in these two layers, peaking at 60–75 ms in layer 4Cα (*Figure 4D*) and at 100 ms in layer 4Cβ (*Figure 4C*). Early, untuned suppression was evident on average in layer 4Cα but was not observed in 4Cβ. Untuned suppression appeared to be a signature of magnocellular-driven cortical layers, as it was pronounced in layers 4Cα and 4B (*Figure 4D, B*).

In layer 2/3, eCRF dynamics predominantly involved a tuned suppression component that peaked at around 100 ms (*Figure 4A*), with little evidence of strong untuned suppression. The timing of tuned suppression differed markedly between the distinct cortical output layers 2/3 and 4B (*Figure 4—figure supplement 1A–D*). The timing of early suppression in layer 6 was similar to that of layer 4Cα. Late tuned suppression in layer 5 was similar to that seen in both layers 4Cα and 4Cβ. Layer 5 neurons showed delayed orthogonal facilitation (*Figure 4E*) that had no counterpart in the input layers, but a larger sample is needed to confirm whether this reflects the emergence of a separate facilitative mechanism within the V1 circuit.

## Relation of eCRF dynamics to input pathways

For each neuron in the supra- and infragranular layers, we measured the similarity in the spatiotemporal dynamics of eCRF modulation with those seen on average in the input layers (4Cα and 4Cβ). We quantified this by cross-correlation of each neuron's eCRF orientation-time map with the average map of all neurons in each input layer as a function of time lag (*Figure 4G–K*). *Figure 4G* shows the cross correlation between layer 2/3 neurons and layer 4Cα or 4Cβ neurons (red and black, respectively). Each trace shows the cross-correlation averaged over all layer 2/3 neurons (solid line, mean; shading, s.e.m.). Layer 2/3 dynamics correlated well with both input layers because all had tuned suppression peaking around 80–100 ms. However, the untuned suppression evident in the input layers was not present on average in layer 2/3, possibly explaining why the correlation between layer 2/3 and 4Cα was slightly weaker than between layer 2/3 and 4Cβ. Layer 4B neurons had a strong correlation with layer 4Cα dynamics (*Figure 4H*). Layer 6 neurons showed a similarly strong correlation with layer 4Cα neurons (*Figure 4K*). By contrast, layer 5 neurons showed similar correlations with both layers 4Cα and 4Cβ (*Figure 4J*), similar to the mixed results in layer 2/3.

There are two hypotheses to explain the situation in which layers 2/3 and 5 match the dynamics of both layers 4Cα and 4Cβ equally well. First, individual neurons may receive mixed input from both streams to their eCRF mechanisms. Alternatively, some neurons in a given layer could be dominated by the 4Cα input whereas others are dominated by 4Cβ input. To address these hypotheses, we directly compared the correlation with layer 4Cα and 4Cβ within individual neurons. This is summarized as the distribution of the difference in correlation values (at zero lag) for each neuron. If individual neurons all received mixed signals from both input streams, then correlation difference would cluster around zero. For layer 2/3 neurons, this does not seem to be the case (*Figure 4L*). There was a broad range of correlation difference values, many negative, indicating layer 4Cβ input, others

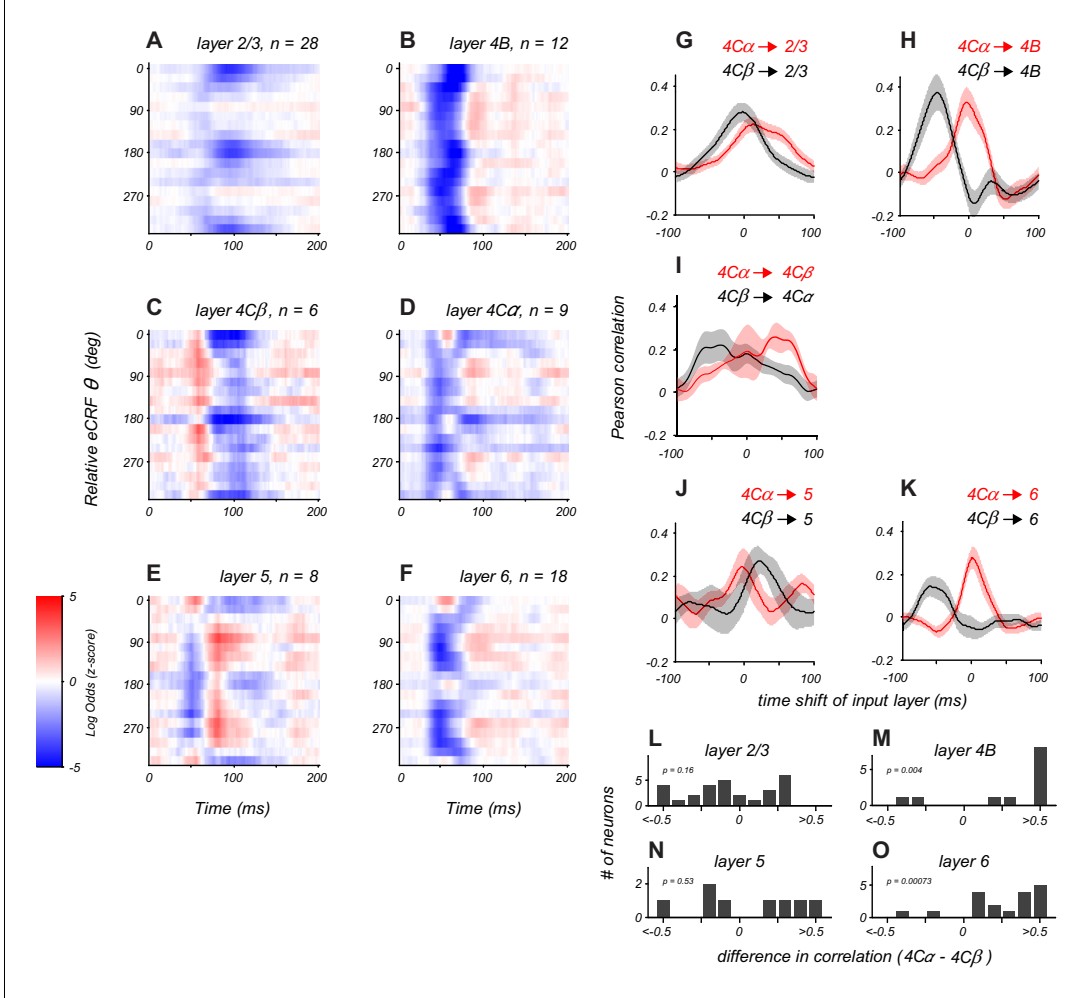

**Figure 4.** Distinct laminar patterns of eCRF modulation. (**A–F**) Average eCRF dynamics by cortical layer for the CRF/eCRF border condition (1x eCRF). Color scale represents the sign and strength of modulation (red, facilitation; blue, suppression). The sample size for each layer is listed above each plot. (**G–K**) Cross-correlation of eCRF kernels with the average kernels of the two divisions of input layer 4C (red, 4Cα; black, 4Cβ) at varying time lags (−100 to +100 ms). Each panel represents the average cross-correlation function for all neurons in a given layer (solid line, mean; shading, s.e.m.). In (**H,K**), the peak at 0 time lag (red) indicates that the eCRF dynamics are well aligned with those of input layer 4Cα; by contrast, the peak at negative time lags (black) indicates that the dynamics of input layer 4Cβ would have to be shifted markedly earlier in time to have peak alignment. Dynamics in layers 2/3 and 5 show comparable agreement with those of both input layers, suggesting less-segregated processing streams. (**L–O**) Histograms show the difference in correlation values with 4Cα and 4Cβ at zero lag for all neurons in each cortical layer.

The online version of this article includes the following source data and figure supplement(s) for figure 4:

**Source data 1.** Source data for *Figure 4*.

**Figure supplement 1.** Patterns of modulation in layers 2/3 and 4B at different eCRF extents.

**Figure supplement 1—source data 1.** Source data for *Figure 4—figure supplement 1*.

positive, indicating a dominant layer 4Cα input, and some around zero, suggesting potential combined input. This suggests that the eCRF of a layer 2/3 neuron does not receive mixed input from layers 4Cα and 4Cβ, but rather receives dominant input from either the M or P pathway. The correlation pattern in layers 4B and 6 was different; there was a clear bias towards positive correlation-difference values, confirming that the majority of individual neurons in layers 4B and 6 (*Figure 4M and O*) had dynamics that were better matched to layer 4Cα than to 4Cβ. Among the eight layer 5 neurons recorded, the correlation difference also split between negative and positive values (*Figure 4N*), suggesting possible stream-specific input onto single neurons.

## Tuning of facilitation and suppression

Previous studies comparing tuning between the eCRF and CRF (*Webb et al., 2005*) involved responses that were integrated over time, and would have engaged multiple distinct eCRF mechanisms. By isolating separate mechanisms of tuned facilitation and suppression, we can directly compare tuning in the CRF and eCRF. By placing the dynamic stimulus within the CRF (to 0.5 x eCRF border), we were able to probe the orientation tuning of facilitation for all neurons rather than that of the restricted population that showed facilitation from the eCRF border. We compared the tuning of the facilitation (arising from extension of the CRF) and tuned suppression (from the eCRF) in individual neurons, by measuring the orientations eliciting peak responses and tuning bandwidths. For each neuron, orientation tuning of facilitation and suppression were averaged over the windows around the peak effect (red and blue bars for the example neuron in *Figure 5A*); timing windows were adjusted on a per-neuron basis.

Orientation tuning was similar for facilitation and tuned suppression. In an example neuron (*Figure 5B–C*), the peak tuning for both mechanisms was around orientations collinear to the neuron's CRF preference. For the population, the peak orientations for tuned facilitation and tuned suppression were within 20 degrees of each other (*Figure 5D, F*). Measures of local selectivity for orientation around the peak (bandwidth) were similarly distributed for both facilitation and suppression (*Figure 5E, G*), although within individual cells, we found only a weak and not statistically significant relationship between the two bandwidths (*Figure 5H*, Pearson's correlation, r = 0.22, p=0.14). More global measures of selectivity (the response ratio at orthogonal to preferred orientations) showed a moderate relationship between the tuning for facilitation and suppression (Pearson's correlation, r = 0.29, p=0.051). Overall, the tuned component of eCRF suppression spanned the same range of selectivity and tuning as that of the facilitation elicited from within and near the CRF, a comparison only made possible by isolating eCRF component mechanisms.

## Modeling of overlapping eCRF mechanisms

The dynamics of eCRF modulation depend on the spatial extent and orientation of the stimulus within the eCRF. Consequently, a given eCRF stimulus may recruit multiple underlying eCRF

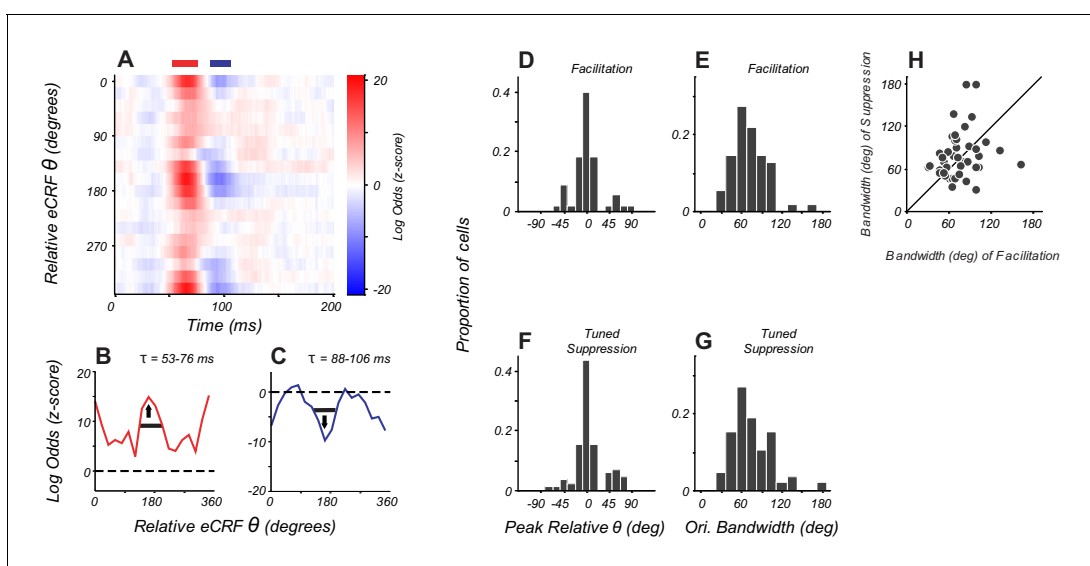

**Figure 5.** Orientation tuning of suppression in eCRF matches CRF tuning. (A) eCRF dynamics from an example cell. Red and blue bars at the top of the figure show the windows used to determine the average tuning. (B, C) Average orientation tuning of eCRF components for the neuron in panel (A). Black horizontal lines indicate orientation bandwidth, arrows indicate peak orientation. (D, F) Population distribution of peak orientations for facilitation and tuned suppression. (E, G) Population distribution of orientation bandwidth (full width at half height) for facilitation and tuned suppression. (H) Relationship between orientation bandwidth for eCRF facilitation and eCRF suppression.

The online version of this article includes the following source data for figure 5:

**Source data 1.** Source data for *Figure 5*.

mechanisms with distinct signs and spatio-temporal profiles. As such, changes in stimulus duration have the potential to alter the net observed steady-state eCRF modulation during stimulus presentation. Briefly presented stimuli will increase the relative influence of short latency eCRF mechanisms; prolonged stimuli will allow both short- and long-latency eCRF mechanisms to modulate CRF responses.

To understand how changes in stimulus properties affect eCRF measurements, we modeled eCRF modulation as a combination of multiple mechanisms that have distinct spatio-temporal dynamics. First, we modeled the response to eCRF stimuli presented collinear with the CRF stimulus, where the eCRF modulation results from a combination of short-latency moderate facilitation and long-latency stronger suppression, akin to the modulation mechanisms that we identified in V1. We simulated a range of stimulus durations from 10 to 1920 ms (*Figure 6A*); for each condition, eCRF modulation was quantified as the average response modulation over the entire stimulus presentation. For brief presentations (<40 ms), there was no net eCRF modulation because both eCRF mechanisms had some intrinsic delay (*Figure 6A*). For intermediate durations (60–120 ms), the model showed net facilitation, because short-latency facilitation was engaged and had greater impact than delayed suppression. For longer durations (>200 ms), stronger long-latency suppression dominated, resulting in net suppressive modulation (*Figure 6A*). Thus, even for a stimulus of fixed orientation in the eCRF, changing stimulus duration altered both the sign and the magnitude of net eCRF modulation.

We also modeled the effect of changing stimulus duration when the eCRF consisted of two separate underlying suppressive mechanisms: a short-latency orientation-untuned and a longer-latency orientation-tuned mechanism (shown by the model eCRF kernel in *Figure 6B*). We measured the steady-state suppression index as a function of eCRF orientation. For stimuli of short-duration, there is no modulation (40–60 ms) (*Figure 6C*). For stimulus durations of 80–100 ms, the measurable eCRF suppression was largely unselective for orientation. For stimulus durations > 200 ms, the response reflected a combination of both orientation-untuned and orientation-tuned mechanisms. As these model responses illustrate, even stimuli of a fixed spatial configuration will produce dramatically distinct net modulations of CRF responses as the stimulus duration is varied and distinct underlying eCRF mechanisms are recruited. The exact manner and degree to which these net eCRF modulations will change with stimulus duration depends upon multiple factors: the underlying mechanisms recruited by a given stimulus, their relative strength or drive, and the comparative differences in their temporal dynamics. In general, however, stimuli of short duration or the integration of spiking activity over windows immediately after stimulus onset will bias net modulation toward component mechanisms that have the shortest latency. By contrast, stimuli of longer duration or integration over time windows that are significantly delayed after stimulus onset will afford more equal contribution by eCRF mechanisms that have diverse temporal dynamics.

## Discussion

In the current study, by probing the dynamics of eCRF modulation, we found three component mechanisms that have distinct timing and spatial profiles: orientation-tuned facilitation, untuned suppression, and tuned suppression. Neurons in different cortical laminae had distinct patterns of eCRF modulation, which were partially segregated with input layer processing streams and were elaborated along distinct corticocortical output pathways. Here, we discuss what these findings can tell us about neural mechanisms and about the role of eCRF modulation in sensory coding.

### Mechanisms of eCRF modulation and link to V1 laminar circuits

#### Tuned facilitation

Facilitation occurred early, was spatially localized to regions within and near the CRF, and generally matched the preferred orientation of the CRF. This eCRF-facilitation component thus probably arises from the same mechanisms that generate visual responses in the CRF. In most cells, facilitation was orientation tuned. Thus we propose that the tuned facilitatory component of eCRF modulation is generated in the compact local circuitry of a few neighboring cortical hypercolumns and, in turn, generates CRF signals.

Some studies have reported facilitation from 'far' regions of the eCRF (*Schwabe et al., 2006*; *Ichida et al., 2007*), but generally only when a low contrast stimulus drove the CRF and when stimuli were absent from regions of the 'near' surround. In our study, there was always a stimulus covering

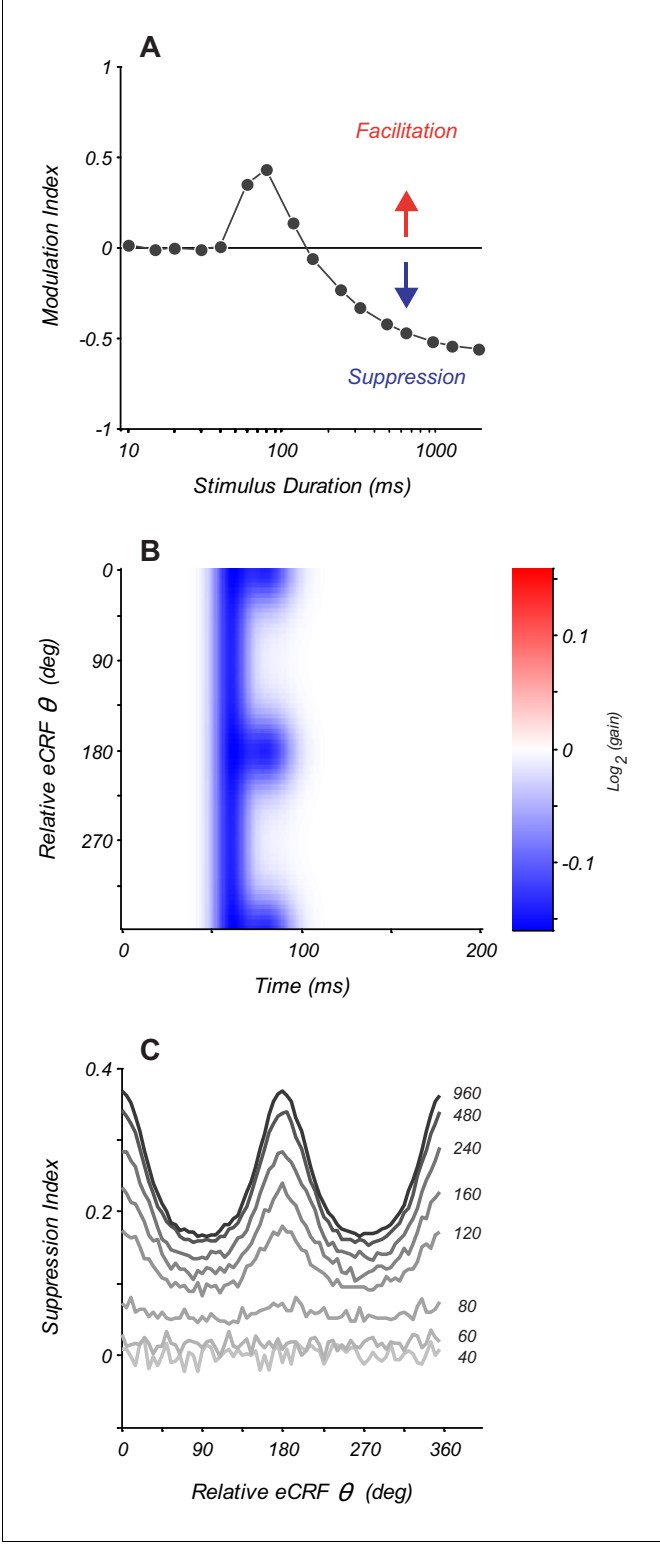

**Figure 6.** Modeling eCRF indicates that net effects change with stimulus duration. (**A**) Response modulation index as a function of stimulus duration for an eCRF model with short-latency facilitation and long-latency strong suppression. Modulation index is calculated as mean change in net spiking activity over the entire stimulus presentation. Positive values reflect facilitation, negative values reflect suppression, and the horizontal line at zero reflects no net modulation. (**B**) eCRF dynamics of a model with both short-latency untuned suppression and long-latency tuned suppression. (**C**) The steady-state suppression index as a function of relative eCRF orientation for
*Figure 6 continued on next page*

*Figure 6 continued*

the model shown in panel (B). Each trace shows a net suppression tuning curve for a stimulus of a fixed duration (duration shown to the right of each curve in ms).

The online version of this article includes the following source data for figure 6:

**Source data 1.** Source data for *Figure 6*.

the near surround, which might explain the lack of facilitation at the largest spatial extents. By contrast, earlier studies that reported strong facilitation (*Kapadia et al., 1995*; *Polat et al., 1998*; *Kapadia et al., 2000*) found facilitation only from flanking stimuli that were relatively close to the CRF. This facilitation is consistent with the spatiotemporal scale of facilitation that we observed and have localized to extensions of a central CRF mechanism. Further, stimuli in these studies were often presented over fairly brief durations, which as our modeling results illustrate will bias net modulation towards this shortest-latency facilitation component.

There was a pronounced facilitation at the orthogonal-to-preferred orientation in the average layer 5 eCRF dynamics (*Figure 4E*). This was also seen, although with a reduced amplitude, in the dynamics from layers 4B, 4C and 6 (*Figure 4B,D,F*). Earlier studies reported examples of orthogonal eCRF facilitation (*Levitt and Lund, 1997*; *Jones et al., 2002*) but the layers of the recorded neurons were not reported. The average dynamics from layers 5 and 6 (*Figure 4E,F*) also showed tuned facilitation at the preferred orientation. The peak time of facilitation at the preferred orientation was 25–50 ms earlier than orthogonal facilitation for the layer 5 kernels (*Figure 4E*), suggesting that different circuits contributed to these two components.

## Untuned suppression

We found that the untuned component of eCRF suppression arose early and was associated with V1 layers receiving or relaying LGN-magnocellular input. This suggests that a proportion of V1 untuned suppression may arise from subcortical eCRF suppression (*Webb et al., 2005*), which has been found in magnocellular–projecting retinal ganglion cells (*Solomon et al., 2006*) and in magnocellular LGN cells (*Solomon et al., 2002*; *Sceniak et al., 2006*; *Alitto and Usrey, 2008*). The short latency, limited spatial scale, and lack of orientation tuning of this component of suppression are consistent with the idea that it derives from feedforward magnocellular LGN input. However, our results suggest that some untuned suppression may arise from local, lateral cortico-cortical inhibition. Although we found that untuned suppression was reduced in strength at distances beyond 2x eCRF border, it was not zero at this larger spatial scale. Broad distributions in the strength and time to peak of this untuned suppression (*Figure 2*) also suggest additional cortical processing beyond common inherited effects from the LGN.

Although untuned suppression was most prevalent in layer 4B, it was also observed in individual neurons in the parvocellular input layer 4Cβ and in layers 2/3 (although it was lost in the average of layer 2/3) and it was found with a delay of 100 ms or more. This delayed component of untuned suppression in neurons with parvocellular-dominated input is likely to be cortically generated, because eCRF suppression is not observed in parvocellular LGN cells (*Alitto and Usrey, 2008*). Together, the evidence suggests that untuned suppression may result from multiple processes, including suppression in magnocellular LGN feedforward input as well as lateral cortico-cortical inhibition.

In a previous study, using steady state stimulation (*Henry et al., 2013*), we reported that for some neurons, the eCRF suppression in layers 2/3 showed components that had high contrast sensitivities, indicative of M-pathway involvement. Furthermore, the suppression was strong at both collinear and orthogonal eCRF orientations, indicating that the untuned component was contributing. In the current results, the main early component is also untuned and we attribute this to the M-pathway, which has characteristically high contrast sensitivity (*Kaplan and Shapley, 1986*). These results refine the view that there are both tuned and untuned components of the eCRF (*Henry et al., 2013*; *Nurminen and Angelucci, 2014*); the untuned component of eCRF suppression may be predominantly derived from magnocellular signals relayed by 4Cα neurons to neurons in other V1 layers.

A recent study on timing of eCRF effects between layers did not appear to show strong M- and P-pathway separation (*Bijanzadeh et al., 2018*). However, it is difficult to compare the results of our current study with those of the Bijanzadeh et al. study, which recorded LFP and MUA using a fixed

size of CRF and eCRF stimuli, because probing for local untuned suppression depends on measuring the border between the CRF and eCRF, which may differ for individual neurons.

## Tuned suppression

Tuned suppression was observed in both input layers and very prominently in layer 2/3. Layer 2/3 suppression peaked at times that were quite delayed compared to those at which facilitation occurred. Both the orientation selectivity and the delayed emergence of this tuned suppression are consistent with its arising as a result of cortico-cortical lateral interactions, as often postulated before (reviewed in *Angelucci et al., 2017*). We found that this suppressive component arises from extended spatial scales, with most of it generated by the activity of V1 neurons with receptive fields more than 2 x the CRF radius away from the recorded neuron. Long-distance V1 lateral connections must be in some way selective for the orientation of their targets in order to support such long-range tuned suppression (*Malach et al., 1993*). If instead this suppression is caused in part by extra-striate feedback (*Angelucci et al., 2002*; *Angelucci et al., 2017*; *Nassi et al., 2013*), then those feedback circuits must also be matched in orientation selectivity to their V1 targets. Tuned suppression is also present in layer 4B, where it differs quantitatively from that of layers 2/3. In layer 4B, tuned suppression is very rapid, and its presence is partially obscured because it overlaps to some extent with untuned suppression (*Figure 4B*). We examined a small sample of layer 4B cells (n = 6) probed using eCRF stimuli with an inner radius of 2x eCRF border, a scale at which the early untuned suppression is no longer present, and found strong early tuned suppression (*Figure 4—figure supplement 1D*). These results point towards a fast, tuned-suppression mechanism that is specific to layer 4B, distinct from that seen in layers 2/3. These results imply that tuned suppression may also be elaborated via multiple circuit mechanisms along distinct cortico-cortical output pathways.

We found that the orientation selectivity of the tuned component of eCRF suppression was similar to that of facilitation arising from regions near the CRF. Yet earlier studies concluded that the tuning of eCRF suppression was broader than that of the CRF (*Webb et al., 2005*). However, these inferences were based on steady-state responses that included both the untuned and tuned components of eCRF suppression and did not discriminate between them.

## Relation to previous work on eCRF modulation

During prolonged visual stimulation with extended patterns, steady-state measurements of eCRF modulation incorporate both facilitatory and suppressive eCRF mechanisms, because their response dynamics are fast compared to stimulus presentation times. In the modeling responses to stimuli with shorter presentation durations, as often used in psychophysical studies, the relative contributions of these mechanisms depended on stimulus presentation time (*Figure 6*). This was due to the fact that the components of facilitation and suppression have distinct temporal profiles. These new results help to reconcile the seemingly contradictory results in the literature.

Studies of eCRF modulation that used large collinear annular gratings drifting for hundreds of milliseconds to many seconds typically reported strong eCRF collinear suppression but little or no collinear facilitation (*Levitt and Lund, 1997*; *Hupé et al., 2001*; *Sceniak et al., 2001*; *Levitt and Lund, 2002*; *Cavanaugh et al., 2002a*; *Jones et al., 2002*; *Shushruth et al., 2012*; *Nassi et al., 2013*; *Henry et al., 2013*; *Trott and Born, 2015*). Other studies that used briefly presented spatially localized stimuli flanking the CRF (*Kapadia et al., 1995*; *Kapadia et al., 2000*) reported a high prevalence of collinear facilitation. In the current study, we observed both types of responses. We found an early facilitation from the defined eCRF border in about 25% of neurons. Most neurons that showed an early facilitation possessed tuned suppression that was temporally delayed (*Figure 3E*). With this biphasic temporal profile, integration of the spiking response over short epochs yields facilitation when compared to CRF stimulation alone (*Figure 6*), as observed by *Kapadia et al., 2000*. Integration of the response over longer stimulus durations leads to net measured suppression as the stronger, delayed suppressive component dominates over facilitation (*Figure 6*), as reported in many studies (*Jones et al., 2001*; *Sceniak et al., 2001*; *Cavanaugh et al., 2002b*; *Webb et al., 2005*; *Hallum and Movshon, 2014*; *Trott and Born, 2015*). We suggest that these differences in temporal integration among multiple eCRF components provide a parsimonious explanation for many previous conflicting reports about eCRF facilitation (and lack thereof).

## Implications for cortical normalization

Through recent work, it has been increasingly argued that normalization represents a 'canonical computation' in neuronal circuits, a computational motif that is repeated in each cortical area to adjust individual neuronal responses on the basis of the average population activity (*Carandini and Heeger, 2011*). Normalization models have proved a useful quantitative framework for characterizing spatial nonlinearities in neuronal responses, ranging from contrast gain control in retinal ganglion cells (*Shapley and Victor, 1979*) to cross-orientation suppression within V1 CRFs (*Carandini et al., 1997*) and V1 eCRF modulation (*Cavanaugh et al., 2002a*; *Cavanaugh et al., 2002b*). Some studies dissect normalization into equivalent specific mechanistic classes, such as 'tuned' normalization (*Verhoef and Maunsell, 2017*; *Ni and Maunsell, 2017*), but often the assumption is that all neurons within a given cortical area possess the same response gating from a single normalization process. As our results in V1 highlight, there are clear differences at the microcircuit level in the scale, tuning and timing of recruited normalization processes. This diversity across neurons is partially explained by laminar differences in overall connectivity with the magnocellular and parvocellular pathways in the cortical input layer. However, even the 'tuned' normalization that we observe in the V1 output layers shows clear distinctions in timing between layers 2/3 and layer 4B, suggesting the existence of further elaborated and segregated microcircuits.

Normalization has been proposed as a computation that serves to adjust neurons' responses to span a non-saturating operating range (*Ringach, 2010*) and to reduce redundancy across the population (*Schwartz and Simoncelli, 2001*). Within this framework, our results suggest that it is incorrect to equate one cortical area with one canonical normalization process common to all neurons. Instead, it may prove more fruitful to dissect the cortical circuit into functional modules based on their downstream targets and to ask what consequences specific normalization mechanisms have for signaling in broader recurrent circuits. In part, this diversity in normalization has been shown to underlie the generation of increasing RF selectivity (*Xing et al., 2005*; *Xing et al., 2011*), resulting in distinct spatiotemporal transformations from the input layer to the various cortical output pathways.

## Implications for perception and models of cortical processing

These findings of multiple component mechanisms of eCRF modulation that have different dynamics lead to interesting predictions for visual perception. They predict that extended stimuli with varied spatial structure will differentially affect the coding of localized stimuli, a phenomenon that is well-characterized in perceptual studies of meta-contrast masking (*Ishikawa et al., 2006*). Further, they predict that stimuli of *identical* spatial structure may influence perception differently, depending on the time frame over which they are viewed. For example, a study that measured discrimination by human and rodent subjects of a central grating patch in the presence of a surrounding patch reported different outcomes between the two species (*Meier and Reinagel, 2013*). There were subtle differences in the temporal presentation of the stimuli in this study. Upon short-duration presentation in the human subjects, surround facilitation was observed, whereas in rats, where there was a longer duration of presentation, there was suppression. The study concluded that there were species differences in eCRF modulation. However, an alternative explanation could be that these modulation differences were due to changes in presentation duration, as shown by our modeling results. With increased stimulus presentation time, the balance shifts from facilitation-dominated to suppression-dominated eCRF modulation. This implies that qualitatively different information about spatial context is being signaled at different points in time following fixation onset, which will dynamically alter the information about the world that is available to organisms to make use of in guiding their behavior.

Statistical models using CRF-eCRF-based neuronal interactions under conditions of natural viewing (*Coen-Cagli et al., 2012*) make a number of predictions about perceptual saliency that match human performance qualitatively. Currently such models are descriptive and static; they do not have any underlying dynamics built in to the receptive field modulation, nor do they allow for potentially different computational goals within the cortical microcircuit. However, cortical processing under natural viewing conditions will engage both spatial and temporal components of receptive fields, partly because of continuous eye-movements (*Rucci and Victor, 2015*). Future models should take into account both the known dynamics of the multiple neural mechanisms that comprise the eCRF and the pathway-specific differences of these dynamics, in order to characterize more precisely

the contextual information relayed from V1 populations to their distinct downstream targets in higher visual cortical areas.

## Materials and methods

### Preparation

Experiments were undertaken on 15 adult male macaque monkeys in compliance with the National Institutes of Health and New York University Animal Use Committee regulations. Detailed procedures have been described previously (*Hawken et al., 1996*; *Ringach et al., 2002*; *Xing et al., 2005*). Single unit recordings were made in anesthetized, paralyzed animals. Initial sedation was induced with ketamine, (5–20 mg/kg, i.m.) followed by initial surgical preparation under isoflurane anesthesia (1–3%). Recording was carried out using sufentanil anesthesia (6–18 µg/kg/h, i.v.) and animals were paralyzed with vercuronium bromide (Norcuron: 0.1 mg/kg/h, i.v.). We continuously monitored heart rate, electrocardiogram, blood pressure, expired $CO_2$, and electroencephalogram to ensure the maintenance of anesthesia and stable physiological state. The pupils were dilated with atropine sulfate (1%), and during the experiment, the eyes were protected by clear, gas-permeable contact lenses and by application of gentamicin sulfate (3%), a topical antibiotic solution. Fixation rings (Duckworth and Kent) were used to minimize any residual eye movements in most experiments and, when used, an ophthalmic anti-inflammatory agent (TobraDex) was also applied.

### Recording and visual stimulation

Single unit recording and recovery of electrode penetrations were the same as recently described (*Henry et al., 2013*). Single unit recordings were carried out by advancing glass-coated tungsten microelectrodes through the cortical layers of V1 to isolate neurons. Action potentials were sorted online, and single-units were isolated using criteria of a fixed spike waveform shape and the absence of spikes during the refractory period. At the end of each recording penetration, 3–5 small electrolytic lesions (2–3 µA for 2–3 s) were made along the length of the recording track. At the end of each experiment, reconstruction of the electrode location of recorded neurons was carried out by locating the electrode track and electrolytic lesions in histological sections. Sections were stained for cytochrome oxidase to reveal clear boundaries between cortical layers, and recording locations were assigned to a precise relative depth within each cortical layer. The assignment method is described in detail in *Hawken et al., 1988*. Only those penetrations that had unambiguous recovery of all lesions and cortical boundaries were used for assignment of cells to cortical layers.

The receptive fields were initially mapped on a tangent screen, eccentricities were between 1 and 6˚. For quantitative studies, stimuli were displayed at a screen resolution of 1024 × 768 pixels and a refresh rate of 100 Hz. The stimuli were presented on either a Sony Trinitron GDM-F520 CRT monitor or an Iiyama HM204DT-A CRT monitor with mean luminances of 90–100 cd/m² and 60 cd/m², respectively. The monitors' luminance was calibrated using a spectroradiometer (Photo Research PR-650) and linearized using a lookup table. Each eye was optimally refracted for the 115 cm monitor viewing distance using external lenses.

### Characterization and determination of CRF and eCRF

Neurons were initially characterized to determine the optimal tuning parameters for orientation, spatial frequency, temporal frequency as in *Henry et al., 2013*. The experiments on surround dynamics described in this paper were conducted in parallel with additional studies not reported here; not all stimuli could be shown for every neuron depending upon recording stability. The eCRF dynamics experiments were often run sooner in the experimental queue in neurons with noticeable eCRF modulation. However, we made sure to study neurons with a diverse range of eCRF properties; the surround suppression indices of our population are comparable to those reported by us previously (*Henry et al., 2013*) and those in earlier studies (*Sceniak et al., 2001*) from our group.

Optimal parameters were used for the stimulus in the CRF in all the experiments performed with the dynamic surround stimulation. Initial experiments were also performed to measure neurons' contrast response functions as well as their size tuning functions to define the boundary between the classical and extra-classical receptive fields (*Henry et al., 2013*). The receptive field center for each neuron was determined by mapping the minimum response field using a small patch of optimal

grating that evoked a response from the neuron (typically ~0.1 degree radius). The size tuning of each neuron was characterized by measuring the change in response as a circular patch of optimal grating stimulus centered over the RF was varied in radius (from 0.1 to >4 degrees). Optimal size was determined to be the smallest radius eliciting peak response. A second experiment to measure the border of the eCRF region consisted of varying the radius of a circular mean gray patch that was centered over a large grating stimulus (square aperture, 8 degrees on a side) over the RF. Increasing the radius of the central gray patch decreased the width of the outer grating region accordingly. This annular tuning experiment was used to determine the region closest to the CRF where surround stimulation still evoked no spiking activity. We defined the eCRF border as the larger of this measurement and the optimal size from the summation experiments.

## Dynamic stimulus

The eCRF stimulus in this study was a dynamic stimulus that was used to map the spatiotemporal impulse-response to oriented gratings in the eCRF. A small patch of drifting sinusoidal grating (of optimal orientation, spatial frequency, and temporal frequency) was presented over the CRF to elicit spiking from the neuron; the contrast of the grating was set to a level eliciting 50% of the maximum response from a neuron in order to be able to detect both increases and decreases in firing that resulted from the surrounding spatial context. Outside of this grating patch (in an abutting annulus), we presented a rapid dynamic stimulus sequence of high-contrast (99%) drifting gratings of changing orientation (*Figure 1A*). The annular gratings in the eCRF had the same spatial frequency as the CRF grating, and consisted of 18 orientations separated by 20 degree steps; each grating was presented for two monitor frames (20 ms at 100 Hz refresh) and its temporal frequency during those two frames was matched to that of the CRF stimulus. The dynamic gratings were presented at one of two spatial phases, either identical to the spatial phase of the coincident central grating or a phase 180 degrees offset from it. Thus, annular gratings collinear to the central grating (when the relative orientation difference was 0 degrees) were either in-phase or out-of-phase with the grating in the CRF (and remained in this relative phase alignment for the 20-ms presentation of the eCRF stimulus); the analysis of eCRF modulation described for most data sets averages over both spatial phases. Occasionally, the annular stimulus contained a screen of mean grey luminance (blank) for two frames instead of a sinusoidal grating; this blank stimulus was included for reference to compare with the influence of surrounding gratings (*Figure 1A*).

The outer edge of the annulus was windowed by a square aperture, typically of a width of 8 degrees of visual angle. We set the radius of the stimulus (the radius of the central grating as well as the inner radius of the dynamic annulus) to be the size of the eCRF boundary (see *Figure 3E*). The eCRF boundary was determined by the smallest radius at which a neuron gives a peak response to a central grating and the smallest inner radius of a grating annulus that elicits no response from the neuron. The larger of these two sizes was defined as the eCRF boundary. For all neurons, we collected data where the border between the central grating and annular grating stimuli was set to the determined eCRF boundary. For a subset of neurons, we ran further experiments in which we presented stimuli with borders that were set to sizes half (*Figure 3D*) and twice that of the eCRF boundary (*Figure 3F*). For example, at smaller sizes, this means that the central drifting grating has a smaller radius and the dynamic annulus also has a smaller inner annulus; they remained spatially abutting. The same applied to the condition in which the eCRF boundary was twice the radius for the border condition. Varying the spatial location of the stimulus border was designed to allow us to compare eCRF modulations at different spatial extents with regard to the CRF-eCRF border.

Visual stimuli (CRF drifting grating and eCRF dynamic sequences) were displayed in 10-s trials, separated by a blank period of mean gray screen for 1 s; 40–50 trials were recorded per experiment. Overall, neuronal firing rates were relatively stable during stimulus presentation, with a response ratio of $0.87 \pm 0.02$ (median ± s.e.m.) for mean firing rates in the last 2 s of a trial compared to the first, and a response ratio of $0.94 \pm 0.02$ for mean firing rates in the last 10 trials compared to the first 10 trials. Thus, although neurons exhibit some changes in firing rate across the experiment, the changes are small and suggest a relatively steady state of contrast adaptation. Our analyses focus on the average dynamics within this state of steady drive; in addition, post hoc sorting of the data by trial or time window within a trial did not yield obvious changes in the results presented here.

## Data analysis

### Reverse-correlation

Analysis of the neural responses consisted of standard subspace reverse-correlation methods (*Ringach et al., 1997*); a schematic illustrating the analysis is shown in *Figure 1A*. From the neuron's spiking activity, we can calculate the probability that a specific orientation in the surround ($\theta$) was presented at a given time ($\tau$) prior to a spike: p($\theta \mid \tau$, spike). Similarly, we can calculate the probability that a mean grey luminance stimulus was on in the eCRF prior to a spike: p(blank $\mid \tau$, spike). The modulation of spiking that results from a specific orientation in the eCRF is calculated as the log odds-ratio of these two probabilities: log [p($\theta \mid \tau$, spike) / p(blank $\mid \tau$, spike)]. Positive values of log odds-ratio indicate that the oriented grating occurred more often on average (at a given time prior to spikes) than a blank stimulus, which we interpret as the eCRF surround grating driving spiking activity (facilitation from the eCRF). Negative values of log odds-ratio indicate that the oriented grating occurred less often on average than a blank, which we interpret as decreases in spiking activity (suppression from the eCRF). Gratings that evoke no response modulation from the eCRF will occur with a probability equal to that of the blank stimulus prior to spiking, and will have log odds-ratios near zero (*Figure 1B*). All of these probabilities are calculated at multiple times prior to spiking, with tau ranging from 0 to 200 ms (*Figure 1D-F*). At extremely short or long values of tau, there will be no effect and the probability of any given stimulus will be equal to that of a blank (*Figure 1B*). At intermediate time scales, there may be modulatory influence that leads to the occurrence of given surround stimuli more or less often than that of a blank (*Figure 1C*). In the example cartoon, there is an increase in probability at the same orientation (collinear) as the CRF stimulus; this would be interpreted as facilitation.

The constant center stimulus and dynamic surround maintain the system in a constant state of adaptation or normalization. Even though the center stimulus is not broadband, it is driving the neuron with a relatively constant rate and the dynamic pattern in the surround is keeping the surround in a constant level of adaptation. Therefore, the timing of the response components is due to small signal perturbation operating in a linear range.

### Quantifying components of eCRF modulation

The LORs gave us the temporal impulse-response modulation produced by a given grating in the eCRF. To determine the statistical significance of these impulse-responses, we normalized the log-odds ratios by transforming them into z-scores by dividing by the standard deviation of the noise in the kernels. The earliest signals that arrive in V1 when a stimulus is presented occur after around 30 ms, due to the inherent latencies in the feed-forward neural circuitry (*Maunsell and Gibson, 1992*). Thus, any measured values that arrive earlier than that are assumed to be due to noise; we used the standard deviation of the LORs (across all stimuli) in the range of 0–20 ms prior to spiking activity as our estimate of the variance in the data and use this to convert the LORs into z-scores. Non-parametric estimation of confidence bounds on the spatio-temporal kernels (via bootstrap resampling) led to similar measures of significance.

We measured multiple attributes of the strength and timing of eCRF modulation, and used these measures to characterize multiple component mechanisms underlying eCRF modulation. By averaging the impulse-responses for orientations near collinear with the central grating (0 ± 20 and 180 ± 20 degrees relative to center), we can measure the strength and timing of both facilitation and suppression; we refer to these as orientation-tuned mechanisms. The magnitude of these mechanisms is the peak z-scored value over the time course of the response (peak positive value for facilitation and peak negative value for suppression). This also gave us the time of the peak response; to calculate response onset and offset, we found the latest time prior to peak and the earliest time after the peak when the response dropped below a z-scored value of 2. In addition, we measured the magnitude and timing for responses that were orthogonal (averaged over orientations 90 ± 20 and 270 ± 20 degrees relative to center); we refer to these as orientation untuned mechanisms.

### Modeling

Modeling of multiple eCRF mechanisms was carried out to illustrate the manner in which stimulus duration influences the average modulation observed in single neurons from surround stimuli. Neural response to an optimal CRF stimulus was modeled as a homogenous Poisson process firing at an

average rate of 60 Hz. Modulation from eCRF stimuli were modeled as multiplicative (divisive) temporal gain changes of this CRF drive, for mechanisms of eCRF facilitation (suppression). Model parameters for defining the temporal properties of eCRF mechanisms (time of peak response, duration) were chosen to approximate those seen on average across our sampled populations. The temporal kernel for facilitation ($K_f$) was a Gaussian profile in time (amplitude, 1.2; mu – time of peak response, 50 ms; s.d., 5 ms); the temporal kernel for suppression ($K_s$) was a Gaussian profile (amplitude, 1.7; mu, 80 ms; s.d., 10 ms). For comparison of mixtures of facilitation and suppression, the total temporal impulse-response from the eCRF ($K_{total}$) was $K_{total} = (1 + K_f) / (1 + K_s)$; the constant of 1 in the numerator (denominator) is equivalent to unitary gain (no change) in the absence of eCRF facilitation (suppression). The duration of eCRF stimuli were sampled from a range of 10–1920 ms. Neuronal responses were modeled by convolving the eCRF kernel ($K_{total}$) with the surround stimulus sequence, and using the time-dependent output to modulate the gain of the Poisson process representing CRF stimulus drive. For each stimulus duration, average eCRF modulation was quantified by taking the mean firing rates over the entire period of stimulus presentation. Modulation index was calculated as ($R_{CRF+eCRF} - R_{CRF}$)/$R_{CRF}$, whereby positive (negative) values indicate eCRF facilitation (suppression).

To illustrate how the average tuning of eCRF suppression is dependent upon stimulus duration when multiple suppressive mechanisms are present in the eCRF, we modeled eCRF suppression as a combination of untuned and tuned components with distinct spatiotemporal profiles. The temporal kernel for untuned suppression ($K_{us}$) was a Gaussian profile (amplitude, 1.1; mu; 60 ms; s.d., 7 ms) and for tuned suppression ($K_{ts}$) was a temporal kernel of longer latency and duration (amplitude, 1.1; mu, 80 ms; s.d., 10 ms). Untuned suppression was equal for all orientations presented in the eCRF; tuned suppression was created by scaling the amplitude $K_{ts}$ by a Von Mises function alpha, ranging in amplitude from 0 to 1 (peak orientation, 0 deg; half-width at half height, 30 deg). Thus, the temporal impulse-response for all suppression from the eCRF ($K_{total}$) was $K_{total} = 1/(1 + K_{us}) * (1 + alpha*K_{ts})$. Stimulus duration was varied as above, and an average suppression index for each stimulus duration and eCRF orientation was calculated as $1 - (R_{CRF+eCRF})/R_{CRF}$.

## Acknowledgements

This work was supported by NIH R01 EY008300 (MJH) and R01 EY001472 (RMS).

## Additional information

### Funding

| Funder | Grant reference number | Author |
| --- | --- | --- |
| National Institutes of Health | EY008300 | Michael J. Hawken |
| National Institutes of Health | EY001472 | Robert M. Shapley |

The funders had no role in study design, data collection and interpretation, or the decision to submit the work for publication.

### Author contributions

Christopher A Henry, Conceptualization, Data curation, Software, Formal analysis, Validation, Investigation, Visualization, Methodology, Writing - original draft, Writing - review and editing; Mehrdad Jazayeri, Conceptualization, Investigation, Methodology, Writing - review and editing; Robert M Shapley, Conceptualization, Supervision, Funding acquisition, Investigation, Methodology, Project administration, Writing - review and editing; Michael J Hawken, Conceptualization, Supervision, Funding acquisition, Investigation, Visualization, Methodology, Writing - original draft, Project administration, Writing - review and editing

### Author ORCIDs

Christopher A Henry (ID) https://orcid.org/0000-0002-1971-5626
Mehrdad Jazayeri (ID) http://orcid.org/0000-0002-9764-6961

## Ethics

Animal experimentation: All procedures received prior approval by the Institutional Animal Care and Use Committee (IACUC) of New York University (protocol 00-1051) and were carried out in strict compliance with the National Institutes of Health's Guide for the Care and Use of Laboratory Animals.

## Decision letter and Author response

Decision letter https://doi.org/10.7554/eLife.54264.sa1
Author response https://doi.org/10.7554/eLife.54264.sa2

# Additional files

## Supplementary files

• Transparent reporting form

## Data availability

Data generated during this study are included in the manuscript and supporting files. Source data files have been provided for Figures 1 through 6, including figure supplements.

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
