## [Decision Letter]

**Acceptance summary:**

We found your manuscript to be a systematic and compelling contribution to the literature on how visual stimuli that do not themselves evoke responses can facilitate or suppress ongoing visually-driven neural activity in cortex. The paper reveals three main mechanisms that operate on different scales in space and times and charts their relative contribution to cells in different layers in primary visual cortex. The results illuminate a complicated field of research and provide clear predictions for cortical normalisation and processing.

**Decision letter after peer review:**

Thank you for submitting your article "Distinct spatiotemporal mechanisms underlie extra-classical receptive field modulation in macaque V1 microcircuits" for consideration by *eLife*. Your article has been reviewed by three peer reviewers, including Kristine Krug as the Reviewing Editor and Reviewer #1, and the evaluation has been overseen by Andrew King as the Senior Editor.

The reviewers have discussed the reviews with one another and the Reviewing Editor has drafted this decision to help you prepare a revised submission.

Summary:

The manuscript "Distinct spatiotemporal mechanisms underlie extra-classical receptive field modulation in macaque V1 microcircuits" is a systematic, very thorough investigation into the different mechanisms that contribute to the eCRF of V1 neurons in primates. This manuscript describes experiments that use reverse correlation to measure the dynamics of surround suppression in V1. Although a few previous experiments inferred some properties of the surround with reverse correlation, this is the first study to measure this directly. The picture that emerges clearly shows several distinct components that operate on different spatio-temporal scales. The quantification is careful and appropriate. The modeling work is also a useful contribution in demonstrating how stimulus duration can affect the sign and magnitude of the net modulatory effect arising from multiple heterogeneous sources. We found the Discussion useful both as a summary of a number of important results in the literature, and in comparison to those reported here.

The results are important because the study of temporal dynamics illuminates multiple sources of facilitation and suppression. Furthermore, recognizing these different components and the differences in their spatial extents helps to reconcile seemingly contradictory results in the literature. This is a solid and compelling contribution to the literature on how visual stimuli that do not themselves evoke responses can facilitate or suppress ongoing visually-driven neural activity.

We have no fundamental concerns about the presented data and analyses. However, we think that some clarifications of the results and methods are required to understand the presented data better and maximise the insights that can be gained from them.

Essential revisions:

1) For the results described in Figures 2 and 3, it would be important to visualise the intersections of all three different mechanisms across all sampled neurons including the non-significant results. The authors describe the proportion of cells with one or commingled properties (for example, suppression only, or facilitation followed by suppression); but it is not clear how many cells had *no* eCRF effect at all. This would help to understand whether there are distinct, identifiable subsets of neurons in V1 based on the different combinations of eCRF mechanisms.

2) Results: we could not find a full description of the inclusion criteria/sampling strategy for the V1 neurons in this study and the selectivity of the neurons studied. This should be part of the current results together with visual field position, receptive field size, stimulus selectivity and what proportion of cells are simple/complex cells of this sample.

3) The method for layer assignment is not described in the Materials and methods, although the Results explicitly refer to the Materials and methods for this. Linked to this, the layer analysis has for some layers a very small n, so the methods of assignment and an understanding of sampling (point 1 above) are important to judge how representative and robust these results are.

4) An explanation of how in the 0.5 x eCRF condition, the authors distinguish eCRF from CRF mechanisms contributing to the measured effects would be useful.

5) Results: Figure 4. The discussion of this figure focusses on correlation values at zero lag (appropriately). But G-J show results at different time lags. These are interesting, but slightly hard to interpret without also seeing what this function looks like comparing 4Cα with 4Cβ. We suggest adding this trace. For Figure 4 H and J: a brief reference of the meaning and significance of the different patterns relative timing of peaks in these results would be helpful.

6) Stability of neural responses: The authors should report the total stimulus duration, i.e. how long was the center only (CRF) stimulus continuously displayed between blank epochs? A related question is what would responses look like if the CRF-only stimulus was presented for that same duration; would responses be stable, or adapt significantly? This is pertinent to the authors' statement that recorded cells were in a constant adaptation state. Clearly, they were in a steady light adaptation state, but cortical cells are known to adapt to contrast. So it is important to document that mean firing rate was stable, or would have been stable across the entire epoch in the absence of the eCRF stimuli. Or did the authors discard cells whose responses were *not* stable?

Were there any differences to late vs. early stimulus conditions, for example if they compared the average first 100 repeats of a given surround flash condition with later/final 100 repeats of same configuration. This again gets at stability of responses. Perhaps this was addressed in a prior paper from the group, but it would be useful to remind readers here.

7) It would be helpful if they were to state in the description of the modeling how they chose the specific values of mu for facilitation, and tuned and untuned suppression. Were they simply taken from population averages of the data in this study? How sensitive are the modeling results to the specific parameter values chosen?

---

## [Author Response]

Essential revisions:1) For the results described in Figures 2 and 3, it would be important to visualise the intersections of all three different mechanisms across all sampled neurons including the non-significant results. The authors describe the proportion of cells with one or commingled properties (for example, suppression only, or facilitation followed by suppression); but it is not clear how many cells had no eCRF effect at all. This would help to understand whether there are distinct, identifiable subsets of neurons in V1 based on the different combinations of eCRF mechanisms.

To clarify how these multiple mechanisms co-occur within individual neurons and across the population as a whole, we have added panel G to Figure 2 and the following text to the Results:

“In individual neurons, the distinct eCRF components were present in varying combinations. […] The pie chart in Figure 2G provides a summary visualization of the percentage of neurons that exhibited each of these specific combinations of eCRF components.”

Here is the added legend for panel 2G:

“(G) Percentage of neurons in the population exhibiting specific combinations of significant eCRF components. […] The absence of significant eCRF modulation is shown in grey.”

2) Results: we could not find a full description of the inclusion criteria/sampling strategy for the V1 neurons in this study and the selectivity of the neurons studied. This should be part of the current results together with visual field position, receptive field size, stimulus selectivity and what proportion of cells are simple / complex cells of this sample.

We have added the following text to the Materials and methods to clarify the sampling strategy for the neurons included in this study:

“The experiments on surround dynamics described in this paper were conducted in parallel with additional studies not reported here; not all stimuli could be shown for every neuron depending upon recording stability. […] However, we made sure to study neurons with a diverse range of eCRF properties; the surround suppression indices of our population are comparable to those reported by us previously (Henry et al., 2013) and in earlier studies (Sceniak et al., 2001) from our group.”

To address the physiological properties of the neurons included in this study, we have added the following text to the Results:

“The time course of eCRF modulation was measured and its dependence upon orientation, spatial phase, and spatial extent determined using a reverse-correlation stimulation paradigm (Figure 1) in 106 V1 neurons (37 simple, 69 complex cells). […] Tuning for preferred stimulus size, measured as the aperture radius of an optimal stimulus, also spanned a wide range with optimal sizes ranging from 0.12 to 4 degrees (median: 0.45 degrees) in agreement with distributions reported in a prior study (Sceniak et al., 2001).”

3) The method for layer assignment is not described in the Materials and methods, although the Results explicitly refer to the Materials and methods for this. Linked to this, the layer analysis has for some layers a very small n, so the methods of assignment and an understanding of sampling (point 1 above) are important to judge how representative and robust these results are.

That is our mistake, thank you very much for pointing this out. The layer assignment is standard in our laboratory and has been described in a number of publications. A new paragraph has been added to the Materials and methods section to describe this, as follows:

“Single unit recordings were carried out by advancing glass-coated tungsten microelectrodes through the cortical layers of V1 to isolate neurons. […] Only those penetrations that had unambiguous recovery of all lesions and cortical boundaries were used for assignment of cells to cortical layers.”

4) An explanation of how in the 0.5 x eCRF condition, the authors distinguish eCRF from CRF mechanisms contributing to the measured effects would be useful.

Thank you for pointing out this need for clarification. Stimuli in the 0.5 x eCRF condition were run (and expected) to possibly encroach upon the CRF mechanisms themselves, as a direct test whether facilitation observed from beyond the 1.0x eCRF border exhibited response profiles akin to those obtained by excitatory drive from the edges of the CRF. We have added the following text in the Results to clarify this point:

“For those neurons that exhibited facilitation at the eCRF border, there was no significant difference in the time of peak facilitation when probed with the closer 0.5 x eCRF border condition (p = 0.61, paired t-test). […] Together, these two results suggest that the facilitative signals from near regions of the eCRF were most likely driven by the spatial continuation of central CRF mechanisms.”

5) Results: Figure 4. The discussion of this figure focusses on correlation values at zero lag (appropriately). But G-J show results at different time lags. These are interesting, but slightly hard to interpret without also seeing what this function looks like comparing 4Cα with 4Cβ. We suggest adding this trace. For Figure 4 H and J: a brief reference of the meaning and significance of the different patterns relative timing of peaks in these results would be helpful.

We have added a panel to Figure 4 showing how the dynamics in a given input layer correlate with those of the other input layer (4Cα -> 4Cβ, and 4Cβ -> 4C). In addition, we have added the following additional sentences to the figure legend to clarify the meaning and significance of these correlation comparisons:

“In (H, K) the peak at 0 time lag (red) indicates the eCRF dynamics are well aligned with those of input layer 4Cα; in contrast, the peak at negative time lags (black) indicates that the dynamics of input layer 4Cβ would have to be shifted markedly earlier in time to have peak alignment. Dynamics in layers 2/3 and 5 show comparable agreement with those of both input layers, suggesting less segregated processing streams.”

6) Stability of neural responses: The authors should report the total stimulus duration, i.e. how long was the center only (CRF) stimulus continuously displayed between blank epochs? A related question is what would responses look like if the CRF-only stimulus was presented for that same duration; would responses be stable, or adapt significantly? This is pertinent to the authors' statement that recorded cells were in a constant adaptation state. Clearly, they were in a steady light adaptation state, but cortical cells are known to adapt to contrast. So it is important to document that mean firing rate was stable, or would have been stable across the entire epoch in the absence of the eCRF stimuli. Or did the authors discard cells whose responses were not stable?Were there any differences to late vs. early stimulus conditions, for example if they compared the average first 100 repeats of a given surround flash condition with later/final 100 repeats of same configuration. This again gets at stability of responses. Perhaps this was addressed in a prior paper from the group, but it would be useful to remind readers here.

We thank the reviewer for raising these relevant points. We have added the following text to the Materials and methods to clarify both the parameters of the stimuli shown (duration, blank) as well as the stability of firing rates throughout each experiment.

“Visual stimuli (CRF drifting grating and eCRF dynamic sequences) were displayed in 10 second trials, separated by a blank period of mean grey screen for 1 second; 40-50 trials were recorded per experiment. […] Our analyses focus on the average dynamics within this state of steady drive; in addition, post hoc sorting of the data by trial or time window within a trial did not yield obvious changes in the results presented here.”

7) It would be helpful if they were to state in the description of the modeling how they chose the specific values of mu for facilitation, and tuned and untuned suppression. Were they simply taken from population averages of the data in this study? How sensitive are the modeling results to the specific parameter values chosen?

We have clarified how the parameter values were chosen by adding the following text to the Materials and methods:

“Modulation from eCRF stimuli were modeled as multiplicative (divisive) temporal gain changes of this CRF drive, for mechanisms of eCRF facilitation (suppression). Model parameters for defining the temporal properties of eCRF mechanisms (time of peak response, duration) were chosen to approximate those seen on average across our sampled populations.”

In addition, we have addressed the generality and robustness of these findings (to changes in the specific parameter values) by adding the following text to the end of the Results section:

“As these model responses illustrate, even stimuli of a fixed spatial configuration will produce dramatically distinct net modulations of CRF responses as the stimulus duration is varied and distinct underlying eCRF mechanisms are recruited. […] In contrast, stimuli of longer duration or integration over time windows significantly delayed after stimulus onset will afford more equal contribution by eCRF mechanisms with diverse temporal dynamics.”